# Comparative neurofilament light chain trajectories in CSF and plasma in autosomal dominant Alzheimer's disease

Anna Hofmann[1,2], Lisa M. Häsler[1,2], Marius Lambert[1,2], Stephan A. Kaeser[1,2], Susanne Gräber-Sultan[1], Ulrike Obermüller[1,2], Elke Kuder-Buletta[1], Christian la Fougere[1,3], Christoph Laske[1,4], Jonathan Vöglein[5,6,7], Johannes Levin[5,6,7], Nick C. Fox[8], Natalie S. Ryan[8], Henrik Zetterberg[9,10], Jorge J. Llibre-Guerra[11], Richard J. Perrin[11,12], Laura Ibanez[11,13,14], Peter R. Schofield[15,16], William S. Brooks[15,17], Gregory S. Day[18], Martin R. Farlow[19], Ricardo F. Allegri[20], Patricio Chrem Mendez[20], Takeshi Ikeuchi[21], Kensaku Kasuga[21], Jae-Hong Lee[22], Jee Hoon Roh[23], Hiroshi Mori[24], Francisco Lopera[25,42], Randall J. Bateman[11], Eric McDade[11], Brian A. Gordon[26], Jasmeer P. Chhatwal[27,28,29], Mathias Jucker[1,2]✉, Stephanie A. Schultz[27,28]✉ & Dominantly Inherited Alzheimer Network*

Disease-modifying therapies for Alzheimer's disease (AD) are likely to be most beneficial when initiated in the presymptomatic phase. To track the benefit of such interventions, fluid biomarkers are of great importance, with neurofilament light chain protein (NfL) showing promise for monitoring neurodegeneration and predicting cognitive outcomes. Here, we update and complement previous findings from the Dominantly Inherited Alzheimer Network Observational Study by using matched cross-sectional and longitudinal cerebrospinal fluid (CSF) and plasma samples from 567 individuals, allowing timely comparative analyses of CSF and blood trajectories across the entire disease spectrum. CSF and plasma trajectories were similar at presymptomatic stages, discriminating mutation carriers from non-carrier controls 10-20 years before the estimated onset of clinical symptoms, depending on the statistical model used. However, after symptom onset the rate of change in CSF NfL continued to increase steadily, whereas the rate of change in plasma NfL leveled off. Both plasma and CSF NfL changes were associated with grey-matter atrophy, but not with Aβ-PET changes, supporting a temporal decoupling of Aβ deposition and neurodegeneration. These observations support NfL in both CSF and blood as an early marker of neurodegeneration but suggest that NfL measured in the CSF may be better suited for monitoring clinical trial outcomes in symptomatic AD patients.

The characteristic pathological features of Alzheimer's disease (AD) are Aβ and tau aggregates. These proteopathic changes are thought to lead to neurodegeneration, which ultimately results in cognitive and functional decline. The detection and monitoring of pathological changes in vivo can be performed by positron emission tomography (PET) and magnetic resonance imaging (MRI) or by analysis of certain proteins in the cerebrospinal fluid (CSF) or blood[1–3].

A full list of affiliations appears at the end of the paper. *A list of authors and their affiliations appears at the end of the paper.

✉e-mail: mathias.jucker@uni-tuebingen.de; saschultz@mgh.harvard.edu

Biomarkers for neurodegeneration are especially important since neurodegeneration is most proximal to the cognitive decline[4,5]. Neurofilament light chain protein (NfL) is a cytoskeletal component that is mainly found in myelinated axons[6]. An increase in NfL in blood and CSF is observed in AD but also in many other neurodegenerative diseases[7–9] and some non-neurodegenerative diseases[10,11]. It is assumed that NfL is released from damaged neurons and then diffuses into the CSF and blood, serving as a biomarker of neuronal damage.

Previous research has shown that NfL increases in blood and CSF starting in the presymptomatic phase of AD and is a good predictor of cognitive decline[8,9,12,13]. The ability of NfL to detect changes before clinical manifestations makes it a promising biomarker for drug response in trials of disease-modifying therapies in presymptomatic phases[3]. Although cross-sectional NfL levels in AD show a robust correlation between blood and CSF NfL levels[14], it is uncertain to what extent CSF and blood NfL trajectories differ over the 2–3 decade-long period of AD pathogenesis. However, the latter is important for understanding the dynamics and significance of this fluid biomarker and to decide whether the more cumbersome and patient-burdening CSF measurement can be replaced by blood measurements.

Here, we used data and biospecimens from the Dominantly Inherited Alzheimer Network (DIAN) that longitudinally examines individuals from families with highly penetrant autosomal dominant AD (ADAD) mutations in the genes for *APP*, *PSEN1*, or *PSEN2*[15]. Family members who do not carry the mutations (non-carriers; NC) serve as a control group. Since the age at symptom onset is consistent for a given pathogenic variant, an estimated years to symptom onset (EYO) can be calculated for each participant (see "Methods" section) allowing data to be normalized and individuals can be staged relative to their EYO. Previously, we reported on serum NfL trajectories in the DIAN cohort[12,16], albeit based on a much smaller number of participants and without longitudinal CSF data. Here, we perform comparative analyses of trajectories of CSF and blood NfL in relation to MRI volume and Aβ-PET changes in a large ADAD sample and examine the distinct features of plasma and CSF NfL across the ADAD continuum.

## Results

### Association between CSF and plasma NfL

For participant characteristics see Table 1 and Supplementary Table 1. Within the mutation carrier (MC) group, there was a positive correlation between plasma and CSF NfL concentrations when using absolute (cross-sectional; $r^2 = 0.71$ and $p < 2e-16$; Fig. 1a) and longitudinal values (rate of change: $r^2 = 0.57$ and $p < 2e-16$; Fig. 1b). However, within the NC group, plasma and CSF NfL concentrations were only moderately to weakly correlated (cross-sectional: $r^2 = 0.30$ and $p = 2.33e-14$; longitudinal: $r^2 = 0.25$ and $p = 5.65e-07$; Fig. 1a, b).

In both MC and NC groups, higher age and lower body mass index (BMI) were associated with higher absolute plasma NfL levels, while higher age and being male were associated with higher absolute CSF NfL levels (Supplementary Table 2). We used AIC model selection to distinguish among a set of possible models including the above potential confounders. The best-fit model for cross-sectional and rate of change in NfL, carrying 99% and 93% of the cumulative model weight, respectively, included baseline age, baseline BMI, and sex as covariates. After accounting for baseline age, baseline BMI, and sex, the association between plasma NfL and CSF NfL cross-sectionally (MC group: $r^2 = 0.76$ and NC group: $r^2 = 0.40$) and longitudinally (MC group: $r^2 = 0.65$ and NC group: $r^2 = 0.39$), particularly in the NC group, remained moderately weak. Full models are reported in Supplementary Table 3. Age, sex, and BMI were included as covariates in all subsequent analyses.

### NfL trajectory over disease progression

Cross-sectionally, NfL levels in CSF and blood in the MC group began to increase, compared to NC group, between 15-25 years prior to expected symptom onset (Fig. 2a, b with the first difference between MC and NC observed at an EYO of −24.6 years for CSF and −18.9 years for plasma; Supplementary Fig. 1a, b). Similarly, when we examined within-person rate of change, we observed increases in NfL, compared to NC group, between 20-25 years prior to expected symptom onset (Fig. 2c, d; with first difference between MC and NC observed at an EYO of −21.1 years for CSF and −21.2 years for plasma; Supplementary Fig. 2c, d). Supplementary analyses using a piece-wise regression method also resulted in similar trajectories with within-person rate of change diverging somewhat earlier compared to absolute levels (Supplementary Fig. 2). Moreover, a bifurcation point for rate of change in plasma NfL occurring at an EYO of +3.59 years for MC group was found (Supplementary Fig. 2c) where the trajectory of plasma NfL appears to plateau or even decrease after this point. While, comparatively, there appears to be a steady linear increase in the rate of change in CSF NfL across the entire disease course (i.e., no bifurcation point observed at later EYOs; Supplementary Fig. 2d). An alternative generalised additive model (GAM) resulted in overall similar trajectories, but a discrimination of MC vs NC was seen only at -10 years prior to EYO (Supplementary Fig. 2e–h).

To examine how within-person rates of change in CSF and plasma NfL levels differ as a function of cognitive status, we categorized the MC group into presymptomatic, converters, and symptomatic subgroups based on their longitudinal Clinical Dementia Rating scale scores (CDR®; see "Methods" section; Fig. 3a, b and Supplementary Table 4). While the stepwise between-group increase in the within-person rate of change in plasma NfL appears to plateau between converter and symptomatic MC stages (Fig. 3a and Supplementary Table 4), there was a stepwise increase in the within-person rate of change in CSF NfL from presymptomatic MC to converter to symptomatic MC stages (Fig. 3b and Supplementary Table 4).

**Table 1 | Background characteristics of the baseline sample**

| Variable | NC, n = 212[a] | MC, n = 355[a] | p-value[b] |
|---|---|---|---|
| Age (yrs) | 36.6 (10.8) | 37.9 (11.0) | 0.209 |
| Sex | | | 0.882 |
| Female | 119 (56%) | 197 (55%) | – |
| Male | 93 (44%) | 158 (45%) | – |
| EYO (yrs) | −11.4 (11.2) | −8.9 (11.0) | 0.011 |
| BMI | 27.9 (6.3) | 27.6 (5.8) | 0.928 |
| MMSE | 29.7 (6.7) | 27.0 (7.5) | 3.337e-09 |
| CDR global | | | 1.282e-17 |
| CDR 0 | 204 (96%) | 226 (64%) | – |
| CDR 0.5 | 8 (3.8%) | 84 (24%) | – |
| CDR 1+ | 0 (0%) | 45 (13%) | – |
| Family mutation | | | 0.291 |
| *APP* | 37 (17%) | 58 (16%) | – |
| *PSEN1* | 154 (73%) | 274 (77%) | – |
| *PSEN2* | 21 (9.9%) | 23 (6.5%) | – |
| Precuneus volume (mm3) | 9710.9 (1295.1) | 9215.2 (1705.0) | 0.001 |
| Precuneus PiB-PET (SUVR) | 1.2 (0.2) | 2.4 (1.4) | 4.720e-38 |
| Plasma NfL (pg/ml) | 5.8 (3.1) | 9.4 (7.7) | 1.398e-09 |
| CSF NfL (pg/ml) | 243.8 (150.5) | 518.3 (542.7) | 1.222e-10 |

The overall *N* is 212 for NC group and 355 for MC, except for precuneus volume (NC = 200 and MC = 327), precuneus PiB-PET (NC = 186 and MC = 284), plasma NfL (NC = 193 and MC = 314), and CSF NfL (NC = 164 and MC = 278). Note that not all parameters could be collected from each participant. Sex was self-reported as male or female.

*MC* mutation carrier, *NC* non-carrier family member, *EYO* estimated years to symptom onset, *MMSE* Mini Mental Status Examination, *BMI* body mass index, *CDR* Clinical Dementia Rating.
[a]Mean (SD), n (%).
[b]Wilcoxon rank-sum test (two-sided); Pearson's $\chi^2$ test (two-sided).

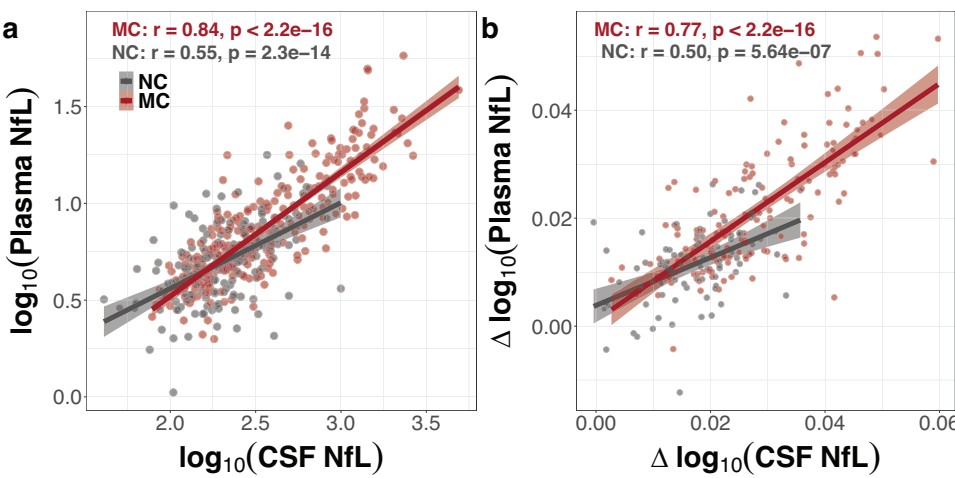

**Fig. 1 | Association between CSF and plasma NfL.** Plasma $\log_{10}$(NfL) levels were associated with CSF $\log_{10}$(NfL) at baseline (**a**) and longitudinally (**b**) in mutation carriers (MC; red) but less so in the non-carriers (NC; grey). There were 274 MC (149 Females; 125 Males) and 162 NC (90 Females; 72 Males) with concurrent baseline CSF and plasma NfL measurements. Within-person annualised rate of change in NfL was extracted from linear mixed effects models (see "Methods" section) for 146 MC and 88 NC with concurrent longitudinal CSF and plasma NfL available. Cross-sectional and longitudinal associations are presented with scatteplot showing unadjusted linear relationship (red and grey bolded lines) between plasma and CSF $\log_{10}$(NfL). The shaded area around each unadjusted linear fit line represents the 95% confidence interval. See Supplementary Table 3 for associations between CSF and plasma log10(NfL) levels after adjusting for baseline age, sex, and baseline BMI.

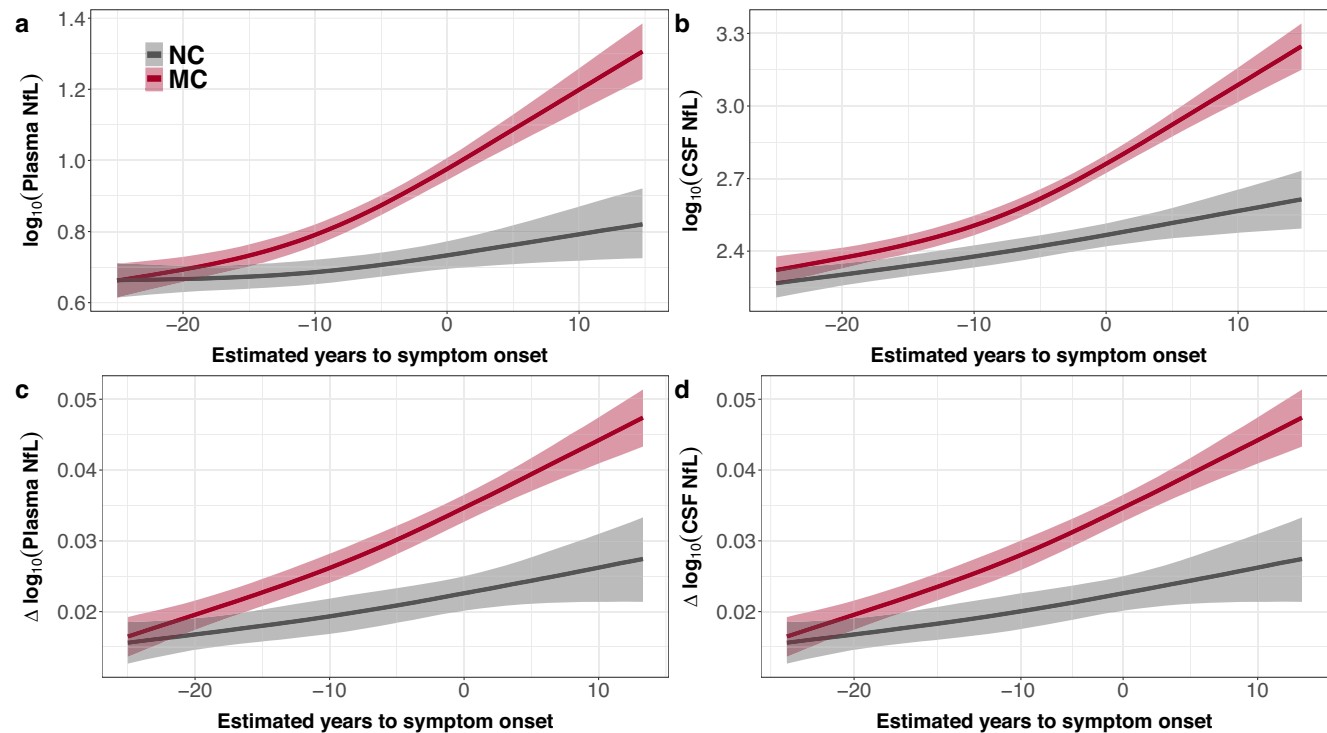

**Fig. 2 | CSF and plasma NfL trajectories across the disease course.** Baseline plasma (**a**) and CSF (**b**) $\log_{10}$(NfL) concentrations in mutation carriers (MC; red) begin to increase, relative to non-carriers (NC; grey), around 10-15 years prior to estimated symptom onset. Similarly, within-person rate of change in plasma (**c**) and CSF (**d**) $\log_{10}$(NfL) levels in MC begin to increase, relative to NC, around 15-20 years prior to estimated symptom onset. The curves and credible intervals are drawn from the actual distributions of model fits derived by the Hamiltonian Markov chain Monte Carlo analyses (see "Methods" section). The shaded areas represent the 95% credible intervals around the model estimates. The first point in the disease course (using estimated years to symptom onset) where NC and MC differed was determined to be the first point where the 95% credible intervals around the difference distribution between NC and MC did not overlap (see Supplementary Figs. 1 and 2 for corresponding longitudinal spaghetti plots and difference distribution plots).

To explore this potential discordance between CSF and plasma NfL further, we examined a ratio of plasma/CSF NfL across these clinical stages. The symptomatic MC group showed a robust decrease in plasma/CSF NfL ratio and could be clearly distinguished from NC and presymptomatic groups (Fig. 3c and Supplementary Table 4).

## NfL trajectory is associated with atrophy rate but less so with amyloid burden

In the symptomatic MC groups, but not the NC or presymptomatic MC groups, the within-person rate of change in NfL in both plasma and CSF was strongly associated with grey matter atrophy (Fig. 4a, b and

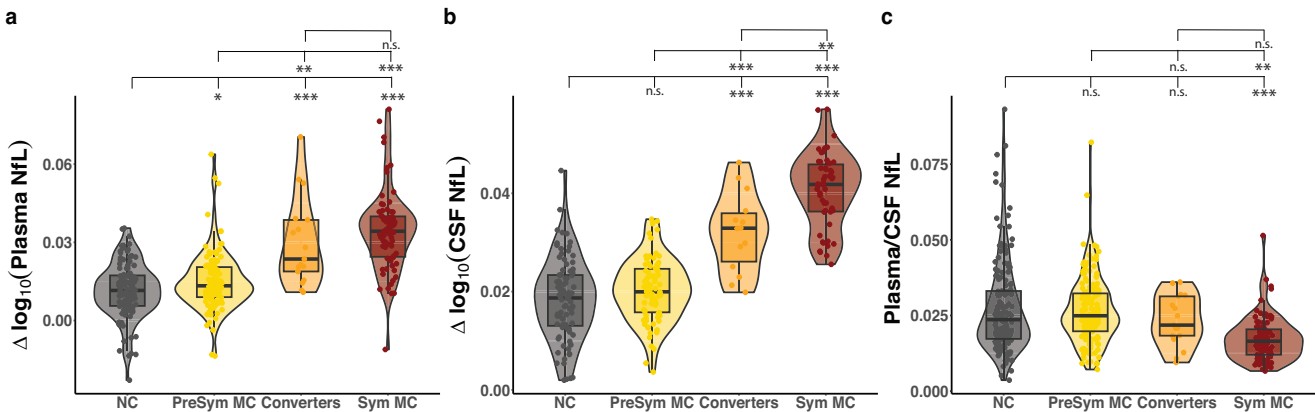

**Fig. 3 | Association between rate of change in NfL and AD clinical groupings.** Rate of change per year in plasma and CSF $\log_{10}$(NfL) across non-carriers (NC; grey, plasma $n = 88$; CSF $n = 89$); Presymptomatic (Presym) mutation carriers (MC; yellow, plasma $n = 79$; CSF $n = 82$; individuals with CDR = 0 across all visits); Converters (orange, plasma $n = 13$; CSF $n = 13$; MC with CDR = 0 at baseline and CDR > 0 at all subsequent visits); Symptomatic (Sym) MC (red, plasma $n = 48$; CSF $n = 50$; MC with CDR > 0 across all visits). (**a**) Annualised rate of change of plasma $\log_{10}$(NfL). Presym MC had a significantly higher rate of change compared to NC. Converters had significantly higher rate of change compared to both NC and presym MC. Sym MC had significantly higher rates of change compared to NC, presym MC and converters. (**b**) Annualised rate of change of CSF $\log_{10}$(NfL). Presym MC had a significantly higher rate of change compared to NC. Converters had significantly

higher rate of change compared to both NC and presym MC. Sym MC had significantly higher rates of change compared to NC, presym MC, and converters. (**c**) Ratio of absolute plasma to CSF NfL levels. Sym MC had a significantly lower ratio compared to NC and presym MC. NC, Presym MC, and Converters had similar plasma/CSF ratios. The boxes map to the median, 25th and 75th quintiles, and the whiskers extend to 1.5 × interquartile range (IQR). The violin plots illustrate kernel probability density (i.e. the width of the shaded area represents the proportion of the data located there). Comparisons were done with linear mixed effects models adjusting for baseline age, baseline BMI, and sex. Corresponding unstandardized beta estimates, standard errors, and multiple comparison corrected exact $p$-values are reported in Supplementary Table 4. n.s. > 0.05; *$p < 0.05$, **$p < 0.01$, ***$p < 0.001$.

Supplementary Table 5). There was no significant association between longitudinal amyloid-PET accumulation and longitudinal CSF or plasma NfL (Fig. 4c, d and Supplementary Table 5).

## Discussion

In AD, the accumulation of cerebral amyloid-β begins decades before clinical onset and is followed by a decline in cortical metabolism and increasing brain atrophy years later[17,18]. Disease-modifying therapies are likely most beneficial when initiated in the presymptomatic phase of disease[3,19]. Therefore, biomarkers for the various preclinical stages suited for eligibility screening of patients and readouts in clinical trials are of utmost importance. Blood-based measurements of NfL have shown promise as a cost-effective, minimally-invasive biomarker to track disease-associated neurodegeneration in certain neurological diseases, including ADAD[12,20,21]. Our prior work has shown NfL to be closely tied to neurodegeneration and white matter integrity[16] and established strong associations between cross-sectional CSF and serum NfL within the ADAD cohort[12]. Here, with a much larger sample size, we find similar associations between NfL measured in CSF and plasma not only for the baseline visits, but also for longitudinal analysis (i.e., within-person rate of change), suggesting elevations in blood NfL may reflect disease-related changes throughout the central nervous system in ADAD, particularly in the presymptomatic stage.

In comparison, within sporadic AD and healthy older adult populations the correlation between NfL levels in CSF and blood is heterogeneous, with some prior reports suggesting strong concordance between NfL levels in the CSF and blood[7,22,23], while others suggesting large unexplained variability[9,24]. In the DIAN NC controls, we observed a weaker correlation between CSF and plasma NfL concentrations compared to the association within MC. Physiologically, both CSF NfL[25,26] and blood NfL[27,28] increase with age, while plasma NfL seems to decrease with increasing BMI[29]. Accounting for baseline age, BMI, and sex improved our models examining the cross-sectional and longitudinal associations between CSF and plasma NfL, in both MC and NC groups. However, there remained about 60% unexplained variability between CSF and plasma NfL in NC controls. While ADAD and sporadic AD share a common pathophysiology and progression,

individuals with sporadic AD are typically older and have more comorbidities, compared to those with ADAD. This complex dependency on systemic factors in sporadic AD will be important to consider in future studies, and recent work supports the use of age- and sex-adjusted reference values for NfL[30,31].

Overall, the CSF and plasma NfL trajectories were found to be very similar at presymptomatic stages and discriminated MC from NC at 10–20 years before EYO, depending on the modeling used. Importantly, however, when examining the rate of change in NfL over the disease course (EYO) and according to cognitive status, the rate of change in plasma NfL appears to plateau at later disease stages in symptomatic MC around two years after estimated symptom onset whereas the rate of change in CSF NfL tends to continue to increase even within symptomatic participants. Similar results have been obtained in Huntington patients with higher fold change in CSF compared with plasma NfL closer to symptom onset[32]. Consistently, we observed a decrease in the ratio of plasma NfL/ CSF NfL within symptomatic MC, which may indicate a hypothetical change in the clearing mechanism of NfL from brain to blood with disease progression. An opposite change in the serum/CSF ratio of NfL in Guillain-Barré Syndrome patients has been interpreted as a peripheral contribution of NfL in this disease[33]. Overall, the underlying biological mechanism for these different dynamics of NfL in CSF and plasma in symptomatic phases of ADAD are not well understood. Assays allowing quantification of NfL in body fluids from the central vs peripheral nervous system source would be very helpful as well as the assessment of changes in renal function. Moreover, it will be important to do similar analyses in sporadic AD.

Lastly, NfL rate of change was associated with grey-matter atrophy rates in symptomatic disease stages but not with the cerebral accumulation of amyloid-β measured via amyloid PET. In line with these findings, a temporal uncoupling of amyloid deposition and neurodegeneration was previously reported: CSF NfL starts to increase after amyloid deposition already reaches an obviously critical (half-maximal) level and afterwards continues to robustly increase while amyloid deposition at later stages reaches a plateau[34]. Such a temporal uncoupling of amyloid deposition and NfL is also in line with the

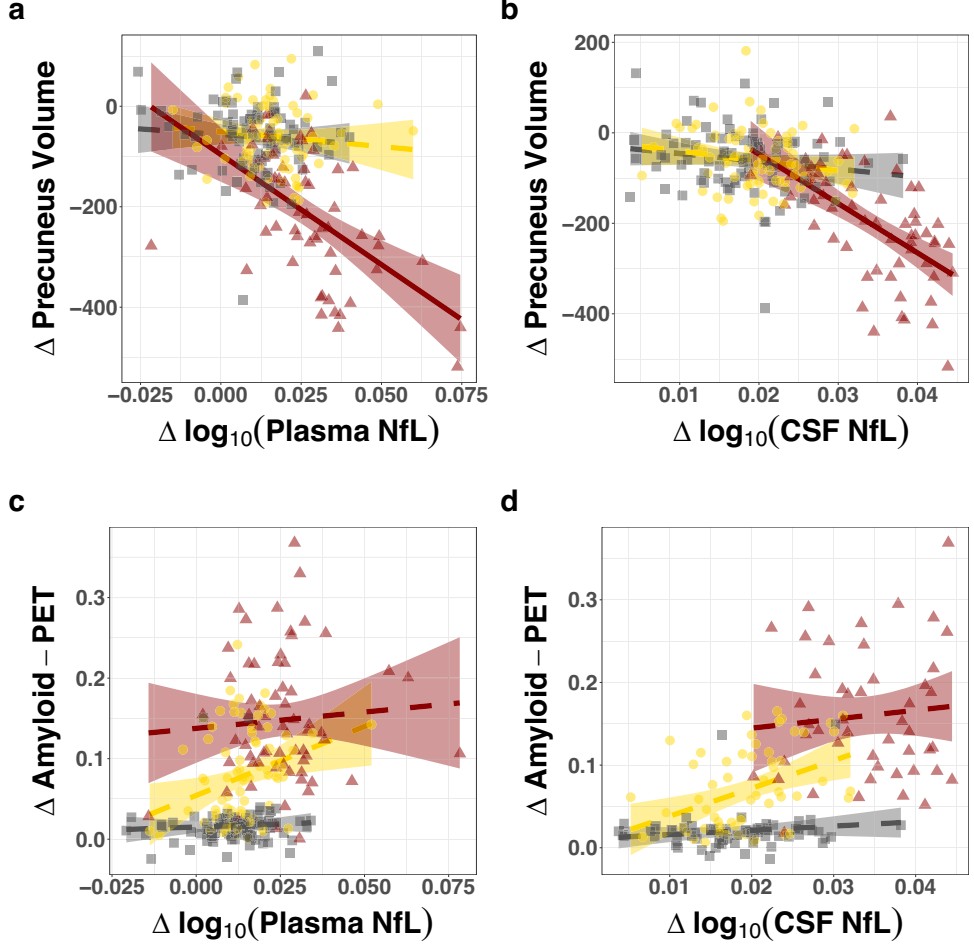

**Fig. 4 | Longitudinal association between NfL, brain atrophy, and amyloid deposition. a**, **b** Relationship between within-person rate of change in $\log_{10}$(NfL) and precuneus grey matter volume for plasma (**a**) and CSF (**b**). Linear mixed effects models were adjusted for baseline age*time, baseline BMI*time, and sex*time. Results revealed a significant association between rate of change in precuneus grey matter and rate of change in plasma and CSF $\log_{10}$(NfL) in the Sym MC group (plasma: $n = 58$ and CSF: $n = 60$), but not in NC (plasma: $n = 81$ and CSF: $n = 82$) or Presym MC (plasma: $n = 76$ and CSF: $n = 78$). **c**, **d** Relationship between within-person rate of change in $\log_{10}$(NfL) and rate of change in precuneus PiB-PET. There were no significant associations between longitudinal PiB-PET and $\log_{10}$(NfL) within

NC (plasma: $n = 89$ and CSF: $n = 65$), Presym MC (plasma: $n = 81$ and CSF: $n = 60$), and Sym MC (plasma: $n = 64$ and CSF: $n = 47$). Shown are non-carrier (NC; grey square), presymptomatic (Presym) mutation carriers (MC; yellow circle), and symptomatic (Sym) MC (including converters to the symptomatic phase, red triangle). The shaded area around each unadjusted linear fit line represents the 95% confidence interval. Solid linear fit line represents a significant association ($p < 0.05$); dashed linear fit line represents non-significant association ($p > 0.05$). Corresponding unstandardized beta estimates, standard errors, and exact $p$-values are reported in Supplementary Table 5. Note that not all participants with longitudinal NfL measurements had imaging parameters available.

absence of a correlation between NfL levels in plasma and amyloid plaque load post-mortem[35].

The strength of the current study is the availability of 567 matched longitudinal CSF and plasma samples, which allowed comparative analyses of the NfL dynamics in both body fluids over the entire disease course, as never before reported. A limitation of the study is that even with this large number or samples, there is limited information at both extremes, i.e. more than 20 years before estimated disease onset and in severely symptomatic patients. Additionally, there are currently relatively few *APP* and *PSEN2* pathogenic variant carriers within in the symptomatic stages of the disease in DIAN, precluding the examination of the trajectories of NfL across different mutation types. As the DIAN study continues to collect longitudinal assessments, future studies will be able to better characterize how these relationships with NfL differ across mutation type and cognitive groups. Another limitation is that the current analysis is restricted to the autosomal dominant form of AD. Although the sporadic and genetic form of AD share similar pathomechanisms, sporadic AD patients are on average 20–30 years older at symptom onset and other age-related comorbidities may impact NfL levels and dynamics[31]. It will be important to do similar

analysis with sporadic AD cohorts, in which most clinical trials are conducted, and to evaluate them in the context of pharmacological interventions.

Nevertheless, the current findings may have implications for ongoing and future clinical AD trials. Currently, NfL is utilized as a readout parameter for therapy success, although with conflicting results[36,37] potentially related to differing disease stages of individuals in the trials[19]. Our findings support the notion that plasma and CSF NfL concentrations are concordant in early disease stages of disease and show similar utility to differentiate early pathophysiology in ADAD MC from NC. However, our observations suggest that in later stages of the disease, CSF NfL changes may reflect the dynamic and ongoing neurodegeneration better than plasma NfL, maybe rendering CSF NfL the better read-out parameter in symptomatic stages of ADAD, compared to plasma NfL. Additionally, our results support previous findings of an uncoupling of amyloid-β deposition and neurodegeneration, beginning in the presymptomatic phases of ADAD, which may partially explain the discordant results of NfL as a biomarker readout across clinical trials that focus on different stages of the disease and AD populations.

## Methods

### Participants

The institutional review board at Washington University in St. Louis provided supervisory review and approval of the multi-site human study (DIAN-Observational study). Each site's institutional review board approved all study procedures. Participants or their caregivers provided informed consent in accordance with their local institutional review boards. Individuals were compensated for their participation. Standardized clinical and imaging assessments were obtained according to The Dominantly Inherited Alzheimer's Network Observational study protocols. Plasma and CSF NfL analysis was approved by the ethics committee at the medical faculty of the University of Tübingen, Germany (project number 718/2014BO2).

DIAN-Observational study (https://dian.wustl.edu/; clinical trial no. NCT00869817) enrolls individuals from families with a known dominantly inherited AD mutation in the *APP*, *PSEN1,* or *PSEN2* genes. Family members who do not carry the mutations serve as controls. We utilized data freeze version 15. DIAN participants are assessed at baseline and subsequent follow-up visits (every second year or annually in case they are symptomatic). Assessment includes the collection of body fluids (CSF and blood), clinical and neuropsychological testing (including Clinical Dementia Rating [CDR®] and Mini-Mental State Examination [MMSE]) and neuroimaging (structural MRI and PET with Pittsburgh Compound B [PiB-PET]) as described below. Sex was self-reported as male or female. The detailed number of participants and samples for baseline and longitudinal measurements are given in Table 1 and Supplementary Table 1.

### Clinical assessment and EYO

The presence of dementia symptoms was assessed using the CDR®[38]. Clinical evaluators were blinded to each participant's mutation status. For every visit a participant's estimated years to symptom onset (EYO) was calculated based on the participant's age at the visit relative to their 'mutation-specific' expected age at symptom onset. The mutation-specific expected age of symptom onset was computed by averaging the reported age of onset across individuals with the same specific mutation[39]. If the mutation-specific expected age at symptom onset was unknown, the EYO was calculated with the estimated age at which the first-degree relative's (parental) cognitive decline began. The parental age of clinical symptom onset was determined by a semi-structured interview with the use of all available historical data. The EYO was calculated identically for both MC and NC. Data from NC with an EYO > 15 years were excluded as there were no corresponding MC with data in this later EYO range and showing these data have the potential for unblinding.

### CDR classification

Presymptomatic MC were defined as individuals who scored CDR (global) = 0 across all visits. Converters are MC who scored CDR = 0 at baseline and CDR > 0 at all subsequent visits. Reverters (those who were CDR > 0 at a visit and CDR = 0 at any subsequent visit) were excluded from analyses. Symptomatic MC are individuals who scored as CDR > 0 across all visits.

### Genotyping

Mutation status was determined using PCR and Sanger sequencing. Individuals with a Dutch-type CAA pathogenic variant (*APP* E693Q; $n = 10$ MC and $n = 11$ NC family members) or variants believed to be non-pathogenic or weakly pathogenic ($n = 13$ and $n = 5$ NC family members) were excluded from this study.

### NfL measurements in the CSF and blood

Blood and CSF samples were collected and initially processed with the same methods described[12]. Specifically, for the current study, all available DIAN plasma samples were shipped to the DIAN site in Tübingen. CSF samples were first shipped to the DIAN site in Munich and used for another analysis before being shipped to the DIAN site in Tübingen. Thus, CSF samples had one additional freeze–thaw cycle in Munich; however, prior work has indicated no significant effect of up to four freeze–thaw cycles on NfL in CSF[40]. For NfL measurement, CSF and plasma samples were thawed on wet ice for one hour. Afterwards they were mixed for 30 s and centrifuged, either briefly (CSF) or for 5 min at 10.000 x g and 4 °C (plasma). Measurements were done on a Single-molecule array platform (Simoa, HD-X analyzer; Quanterix) with commercially available assay kits (NF-Light Advantage Kit Cat 103186). All samples were measured in duplicates. Plasma samples were 1:4 auto-diluted with Simoa NfL sample diluent. CSF samples were diluted 1:100 with Simoa NfL sample diluent before analysis. Inter-assay variability was evaluated with three specific human CSF samples. All samples were measured blinded.

Remeasurement of a subset of the CSF samples was done in 24 samples (2.4%). In 17 of the CSF samples the coefficient of variation (CV) was > 20% and in seven of the samples only one technical replicate measurement was obtained. The remeasured values were taken for the analysis. For plasma, 20 samples had a CV > 20% (max. 36%) and 8 had only one technical replicate. However, we did not perform a remeasurement on these 28 samples (2.2%), as there was insufficient fluid to perform assay. Thus, the initial measurement values were included in the analysis.

Visual inspection of longitudinal CSF and plasma NfL identified two NC and one MC who had extreme values for CSF and plasma NfL for their given mutation status and/or EYO. Previous medical histories revealed that these three individuals had competing neurological disorders and were excluded from analyses (to maintain blinding the specific causes are not mentioned here).

### Imaging

A detailed description of the imaging in DIAN has been published[41]. In brief, MRI Imaging data was screened for protocol compliance and artifacts. All sites used a 3 T scanner that was qualified for use at study initiation and was required to pass regular quality control assessments. Volumetric T1-weighted images were acquired for all participants and were processed using FreeSurfer v 5.3 (http://surfer.nmr.mgh.harvard.edu/) and the Desikan-Killany atlas to produce regional estimates of grey matter volume within brain regions. As has been done previously, analyses focused on the precuneus as the a priori region of interest (ROI). Precuneus volumes were adjusted for total intracranial volume prior to statistical analysis.

Amyloid PET imaging was performed with a bolus injection of ~15 mCi of [$^{11}$C] PiB. Dynamic acquisition consisted of either a 70-min scan starting at injection or a 30-min scan beginning 40 min post injection. For analysis, the PiB-PET data in the common time frame between 40–70 min was used. Using FreeSurfer ROIs, standardized uptake value ratios (SUVRs) were calculated using the cerebellar grey matter as a reference region (PET Unified Pipeline, https://github.com/ysu001/PUP). To minimize the impact of partial volume effects on the PET signal, an RSF-based approach for partial volume correction was used for all regional PET measurements[42]. We chose a single precuneus (average of both hemispheres) region, based on prior work in this and other ADAD cohorts indicating this region as the earliest effected by a number of different imaging measures.

### Statistical analyses

### Relating baseline CSF and plasma NfL

Absolute levels of plasma and CSF NfL were non-normally distributed. Log$_{10}$-transformation was done prior to all analyses, unless otherwise noted. The relationship between baseline CSF and plasma NfL was determined by using Pearson's Correlation and linear regression models implemented in R including covariates for age, sex, and body mass index (BMI). Separate models were fitted for NC and MC. AIC

model selection was performed to distinguish among a set of possible models including the potential confounders of age, sex, and BMI. The best-fit model was determined by cumulative model weight.

### NfL trajectory over disease progression

The relationship between EYO and baseline CSF and plasma NfL values was estimated using linear mixed effects models (LMMs). As previously done, to account for potential non-linear effects, EYO was modeled as a restricted cubic spline with knots at the 0.10, 0.50, and 0.90 quantiles. The LMMs for the baseline NfL values (CSF or plasma) included: fixed effects for mutation status; the linear EYO component; the cubic EYO component; the linear EYO by mutation status interaction; the cubic EYO by mutation status interaction; age, sex, BMI, and a random intercept for family. Model parameters were estimated using an open-source package for Hamiltonian Markov chain Monte Carlo analyses, Stan (http://mc-stan.org/) implemented using R. This resampling approach leads to a distribution of parameter estimates across iterations. From this distribution it is possible to estimate the 95% credible intervals of the model fits at every EYO for NC, MC, and the distribution of the difference between NC and MC. The first EYO where groups (NC and MC) differed was determined to be the first point where the 95% credible intervals around the differences distribution between NC and MC did not overlap 0.

Longitudinal data were modeled using similar LMMs. The rate of change in CSF and plasma NfL for each individual was modeled using an LMM with fixed effects of time from baseline (in years) and a random intercept for family, as well as random slope and intercept terms for each participant. The rate of change in NfL for each individual was extracted from the model estimates for subsequent analyses. This model was also used for generating the rate of change for precuneus grey-matter volume and PiB-PET, for each individual. As with the cross-sectional estimates, the relationship between EYO and rate of change in CSF and plasma NfL was estimated using an LMM. The EYO was modeled as a restricted cubic spline with knots at the 0.10, 0.50, and 0.90 quantiles.

Additionally, piece-wise regressions (segmented package in R) were fitted to examine estimated bifurcation points in the trajectories of cross-sectional and longitudinal log10 plasma and CSF NfL across the EYO spectrum for MC and NC groups. Furthermore, an alternative approach using generalized additive models (GAMs; mgcv package in R)[43,44], to account for non-linear relationships between NfL and EYO, were fitted for cross-sectional and longitudinal log10 plasma and CSF NfL as a function of EYO between MC and NC groups.

To determine whether the extracted rate of change in CSF and plasma NfL was significantly different across mutation status and cognitive status we categorized mutation carriers based on cognitive status, where presymptomatic mutation carriers were individuals who scored as CDR = 0 across all visits ($n = 65$), converters were mutation carriers who scored as CDR = 0 at baseline and CDR > 0 at all subsequent visits ($n = 13$), and symptomatic mutation carriers were individuals who scored as CDR > 0 across all visits ($n = 55$). We used LMMs, including a random intercept for family and fixed effects for baseline age, sex, baseline BMI, and group (that is, NC, presymptomatic MC, converters, or symptomatic MC), where group was the term of interest, and the extracted rate of change in CSF or plasma NfL was the dependent variable. Models were computed using lme4 in R.

A ratio of plasma NfL/CSF NfL was generated to explore the potential within-person discordance between CSF and plasma NfL further. This ratio was examined across clinical stages. Absolute NfL values, instead of log-transformed or rate-of-change NfL values, were used to aid in interpretation of the ratio.

### Relating NfL rate of change to imaging rate of change

The longitudinal relationship between the rate of change in CSF and plasma NfL and concurrent rate of change in grey matter volume and

Aβ accumulation was determined within NC and MC (within pre-symptomatic MC and symptomatic MC subgroups). Therefore, separate models were run for each NC, all MC, presymptomatic MC, and symptomatic MC groups. The dependent term for each model was a time-varying imaging biomarker with fixed effect terms for baseline age*time, sex*time, baseline BMI*time, and interaction between extracted rate of change in CSF or plasma NfL and time. Models contained random slope and intercept terms for participants and random intercepts for family. The primary term of interest was the interaction between the rate of change in CSF or plasma NfL and time. Models were fitted using lme4 in R. The unstandardized regression coefficients (B), standard error of the mean (SE), and *P* values from the LMMs and linear regression models are reported in the supplementary tables.

### Reporting summary

Further information on research design is available in the Nature Portfolio Reporting Summary linked to this article.

## Data availability

Individual-level data from the Dominantly Inherited Alzheimer's Network observational study (DIAN-Observational study) cannot be shared publicly owing to the need for participant anonymity. However, DIAN-Observational study data included in this analysis can be accessed by qualified researchers upon request submitted at https://dian.wustl.edu/our-research/for-investigators/dian-observational-study-investigator-resources/data-request-form/.

## Code availability

All analyses were performed in R. Source code available, with publication, at https://github.com/stephaschultz/DIAN_plasma_CSF_NfL[45].

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

## Acknowledgements

We acknowledge the altruism of the participants and their families and contributions of the DIAN research and support staff at each of the participating sites for their contributions to this study. We also extend our gratitude to the staff and institutions that played a pivotal role in this research. We thank Lina Maria Serna-Higuita (Tübingen) and Nicole McKay (Washington University in St. Louis) for help with the statistical analyses. We further would like to thank Estrella Morenas-Rodríguez and Christian Haass (DZNE Munich) for their support regarding the handling of DIAN samples. Funding for data collection and sharing within this project were provided by the Dominantly Inherited Alzheimer Network (DIAN, U19AG032438), funded by the National Institute on Aging (NIA), Alzheimer's Association (SG-20-690363-DIAN), German Center for Neurodegenerative Diseases (DZNE), Raul Carrea Institute for Neurological Research (FLENI), Research and Development Grants for Dementia from Japan Agency for Medical Research and Development (AMED JP23dk0207066, JP22dk0207049), Korea Health Technology R&D Project through the Korea Health Industry Development Institute (KHIDI), Korea Dementia Research Center (KDRC), funded by the Ministry of Health & Welfare and Ministry of Science and ICT, Republic of Korea (RS-2024-00344521), and the Spanish Institute of Health Carlos III (ISCIII). Contributing authors of this manuscript would like to acknowledge training and research support from the Alzheimer's Association (AARF-21-846786, S.A.S.) and the US National Institutes of Health (K01AG084816, S.A.S.).

## Author contributions

This manuscript has been reviewed by DIAN study investigators for scientific content and for consistency of data interpretation with previous DIAN study publications. All included listed authors have substantially contributed to the work presented in this manuscript and have approved the submitted version. Every author agrees to be accountable for their contributions, as well as the contributions of coauthors, to this work, and together they are committed to ensuring a timely resolution for any questions regarding the accuracy or integrity of the included material. The conceptual design was done by M.J. and S.A.S. Experimental work has been performed by A.H. and L.M.H., with the help of M.L. and S.A.K. Data analyses and data visualizations were done by A.H., S.A.K., M.J., and S.A.S. Manuscript writing was done by A.H., M.J., and S.A.S. All other authors (L.M.H., M.L., S.A.K., S.GS., U.O., E.KB., C.LaF., C.Las., J.V., J.L., N.C.F., N.S.R., H.Z., J.J.LG., R.J.P., L.I., P.R.S., W.S.B., G.S.D., M.R.F., R.F.A., P.CM., T.I., K.K., JH.L., JH.R., H.M., F.L., R.J.B., E.M., B.A.G., and J.P.C.) made contributions to the collection of biospecimens and/or manuscript writing and editing.

## Funding

## Competing interests

R.J.B. receives research funding from the US NIH, Biogen, AbbVie, Bristol Myers Squibb, Novartis, the US National Intelligence Authority, US National Institute of Neurological Disorders and Stroke, Centene, the Rainwater Foundation, the BrightFocus Foundation, the Association for Frontotemporal Degeneration Biomarkers Initiative, Coins for Alzheimer's Research Trust Fund, the Good Ventures Foundation, Hoffman–La Roche, CogState, Signant, the Cure Alzheimer's Research Trust Fund, Eisai, and C2N Diagnostics; receives royalties or licenses from C2N Diagnostics payment or honoraria for lectures, presentations, speakers bureaus, manuscript writing, or educational events from the Korean Dementia Association, the American Neurological Association, Fondazione Prada, Weill Cornell Medical College, Harvard University, Beeson, and Adler Symposium. G.S.D. receives research funding from the US NIH, the Alzheimer's Association, and the Chan–Zuckerberg Initiative; consulting fees from Parabon Nanolabs and Arialysis Therapeutics; and payment or honoraria for lectures, presentations, speakers bureaus, manuscript writing, or educational events from PeerView Media, Continuing Education, Eli Lilly, DynaMed, and SixSense Concierge. E.M. receives research funding from the US National Intelligence Authority, Eisai, Eli Lilly, Roche, and the Gerald and Henrietta Rauenhorst Foundation; consulting fees from AstraZeneca, Sanofi, and Merck; and payment or honoraria for lectures, presentations, speakers bureaus, manuscript writing, or educational events from the Alzheimer Association, Projects in Knowledge, and Neurology Live. H.Z. has served at scientific advisory boards and/or as a consultant for Abbvie, Acumen, Alector, Alzinova, ALZPath, Amylyx, Annexon, Apellis, Artery Therapeutics, AZTherapies, Cognito Therapeutics, CogRx, Denali, Eisai, Merry Life, Nervgen, Novo Nordisk, Optoceutics, Passage Bio, Pinteon Therapeutics, Prothena, Red Abbey Labs, reMYND, Roche, Samumed, Siemens Healthineers, Triplet Therapeutics, and Wave, has given lectures in symposia sponsored by Alzecure, Biogen, Cellectricon, Fujirebio, Lilly, Novo Nordisk, and Roche, and is a co-founder of Brain Biomarker Solutions in Gothenburg AB (BBS), which is a part of the GU Ventures Incubator Program (outside submitted work). J.L. is a consultant for and receives grants, contracts, and royalties from Eisai, Eli Lilly, the German Center of Neurodegenerative Diseases, the German Ministry for Research and Education, the Anton and Petra Ehrmann Foundation, the Luneburg Foundation, Innovationsfonds, the Michael J Fox Foundation, CurePSP, the Jerome LeJeune Foundation, the Alzheimer Forschungs Initiative, Deutsche Stiftung Down Syndrom, Else Kroner Fresenius Stiftung, and MODAG. JLlG receives research funding from the NIH-NIA, the Alzheimer's Association, the Michael J. Fox Foundation, the Foundation for Barnes-Jewish Hospital and the McDonnell Academy. M.J. receives payment or honoraria for lectures, presentations, speakers, bureaus, manuscript writing, or educational events from Eisai. All other authors declare no competing interests.

## Additional information

[1]German Center for Neurodegenerative Diseases (DZNE), Tübingen, Germany. [2]Department of Cellular Neurology, Hertie Institute for Clinical Brain Research, University of Tübingen, Tübingen, Germany. [3]Nuclear Medicine and Clinical Molecular Imaging, University Hospital Tübingen, Tübingen, Germany. [4]Section for Dementia Research, Hertie Institute for Clinical Brain Research and Department of Psychiatry and Psychotherapy, University of Tübingen, Tübingen, Germany. [5]German Center for Neurodegenerative Diseases (DZNE), Munich, Germany. [6]Department of Neurology, Ludwig Maximilians-Universität München, Munich, Germany. [7]Munich Cluster of Systems Neurology (SyNergy), Munich, Germany. [8]Dementia Research Centre, UCL Queen Square Institute of Neurology, London, UK. [9]Department Department of Psychiatry and Neurochemistry, Institute of Neuroscience and Physiology, the Sahlgrenska Academy at the University of Gothenburg, Mölndal, Sweden. [10]Clinical Neurochemistry Laboratory, Sahlgrenska University Hospital, Mölndal, Sweden. [11]Department of Neurology, Washington University School of Medicine, St. Louis, MO, USA. [12]Department of Pathology and Immunology, Washington University School of Medicine, St. Louis, MO, USA. [13]Department of Psychiatry, Washington University School of Medicine, St. Louis, MO, USA. [14]NeuroGenomics and Informatics Center, Washington University School of Medicine, St. Louis, MO, USA. [15]Neuroscience Research Australia, Randwick, NSW, Australia. [16]School of Biomedical Sciences, Faculty of

Medicine and Health, University of New South Wales, Sydney, Australia. [17]School of Clinical Medicine, Faculty of Medicine and Health Sydney, University of New South Wales, Sydney, Australia. [18]Department of Neurology, Mayo Clinic in Florida, Jacksonville, FL, USA. [19]Indiana Alzheimer Disease Center and Department of Pathology and Laboratory Medicine, Indiana University School of Medicine, Indianapolis, IN, USA. [20]Instituto Neurológico FLENI, Buenos Aires, Argentina. [21]Brain Research Institute, Niigata University, Niigata, Japan. [22]Department of Neurology, University of Ulsan College of Medicine, Asan Medical Center, Seoul, Korea. [23]Departments of Neurology and Physiology, Korea University Anam Hospital, Korea University College of Medicine, Seoul, South Korea. [24]Faculty of Medicine, Osaka Metropolitan University, Nagaoka Sutoku University, Osaka, Japan. [25]Grupo de Neurociencias de Antioquia (GNA), Facultad de Medicina, Universidad de Antioquia, Medellín, Colombia. [26]Department of Radiology, Washington University School of Medicine, St. Louis, MO, USA. [27]Department of Neurology, Harvard Medical School, Boston, MA, USA. [28]Massachusetts General Hospital, Boston, MA, USA. [29]Brigham and Women's Hospital Boston, Boston, MA, USA. [42]Deceased: Francisco Lopera. ✉e-mail: mathias.jucker@uni-tuebingen.de; saschultz@mgh.harvard.edu

## Dominantly Inherited Alzheimer Network

David Aguillon[25], Ricardo F. Allegri [20], Andrew J. Aschenbrenner[11], Bryce Baker[11], Nicolas Barthelemy[11], Randall Bateman[11], Jacob A. Bechara[15], Tammie Benzinger[26], Sarah B. Berman[30], William S. Brooks [15,17], David M. Cash[8], Allison Chen[26], Charles Chen[26], Jasmeer P. Chhatwal Chhatwal[27,28,29], Patricio Chrem Mendez[20], Laura Courtney[31], Carlos Cruchaga[13], Alisha J. Daniels[31], Gregory S. Day[18], Anne M. Fagan[11], Martin Farlow[7], Shaney Flores[26], Nick C. Fox [8], Erin Franklin[12], Alison M. Goate[32], Brian A. Gordon [26], Susanne Graber-Sultan[31], Neill R. Graff-Radford[18], Emily Gremminger[31], Jason Hassenstab[11], Elizabeth Herries[11], Anna Hofmann[1,2], David M. Holtzman[11], Russ Hornbeck[26], Edward D. Huey[33], Laura Ibanez[11,13,14], Takeshi Ikeuchi [21], Snezana Ikonomovic[30], Kelley Jackson[31], Steve Jarman[31], Gina Jerome[31], Erik C. B. Johnson[34], Nelly Joseph-Mathurin[26], Mathias Jucker [1,2]✉, Celeste M. Karch[13], Kensaku Kasuga [21], Sarah Keefe[26], Deborah Koudelis[31], Elke Kuder-Buletta[1], Christoph Laske[1,2], Jae-Hong Lee[22], Yudy Milena Leon[25], Allan I. Levey[34], Johannes Levin [5,6,7], Yan Li[32], Jorge J. Llibre-Guerra [11], Francisco Lopera [25,42], Ruijin Lu[31], Jacob Marsh[31], Ralph Martins[35], Parinaz Massoumzadeh[31], Colin Masters[36], Austin McCullough[26], Eric McDade[11], Nicole McKay[26], Matthew Minton[31], Hiroshi Mori[24], John C. Morris[31], Neelesh K. Nadkarni[30], Joyce Nicklaus[26], Yoshiki Niimi[37], James M. Noble[38], Ulrike Obermueller[1,2], Richard J. Perrin [11,12], Danielle M. Picarello[32], Christine Pulizos[31], Laura Ramirez[25], Alan E. Renton[32], John Ringman[39], Jacqueline Rizzo[31], Yvonne Roedenbeck[5,6,7], Jee Hoon Roh [23], Pedro Rosa-Neto[40], Natalie S. Ryan[8], Edita Sabaredzovic[31], Stephen Salloway[33], Raquel Sanchez-Valle[41], Peter R. Schofield [15,16], Jalen Scott[31], Nicholas T. Seyfried[34], Ashlee Simmons[31], Jennifer Smith[11], Hunter Smith[31], Jennifer Stauber[11], Sarah Stout[11], Charlene Supnet-Bell[31], Ezequiel Surace[20], Silvia Vazquez[20], Jonathan Vöglein[5,6,7], Guoqiao Wang[31], Qing Wang[26], Chengie Xiong[31], Xiong Xu[31] & Jinbin Xu[26]

[30]University of Pittsburgh, Pittsburgh, PA, USA. [31]Washington University in St. Louis, School of Medicine, St. Louis, MO, USA. [32]Dept. of Genetics & Genomic Sciences, Dept. of Neuroscience, Ronald M. Loeb Center for Alzheimer's disease, Icahn School of Medicine at Mount Sinai, Mount Sinai, NY, USA. [33]Memory and Aging Program, Butler Hospital, Departments of Psychiatry and Human Behavior and Neurology, Alpert Medical School, Brown University, Providence, RI, USA. [34]Goizueta Alzheimer's Disease Research Center, Emory University, Atlanta, GA, USA. [35]Edith Cowan University, Joondalup, Australia. [36]Florey Institute, The University of Melbourne, Melbourne, Australia. [37]Specially appointed lecturer, Unit for Early and Exploratory Clinical Development, The University of Tokyo, Tokyo, Japan. [38]Taub Institute for Research on Alzheimer's Disease and the Aging Brain, G.H. Sergievsky Center, Department of Neurology, Columbia University Irving Medical Center, New York, NY, USA. [39]Department of Neurology, Keck School of Medicine of USC, University of Southern California, Los Angeles, CA, USA. [40]McGill University, Montreal, Canada. [41]Hospital Clínic de Barcelona. FRCB-IDIBAPS. University of Barcelona, Barcelona, Spain.

