## [Peer Review File · Nature Communications]

Comparative neurofilament light chain trajectories in CSF and plasma in autosomal dominant Alzheimer's diseaseREVIEWER COMMENTS

Reviewer #1 (Remarks to the Author):

This work uses data from the DIAN-Obs cohort to establish associations between plasma and CSF NfL levels across the Alzheimer's disease continuum. They focus on the comparison in the NfL trajectories between fluids (plasma Vs CSF) to elucidate if there are differences between them, information that can be useful for future clinical trials. Of note, a large number of longitudinal samples are included in this study. As some variability in the NfL levels between subjects is still not explained due to the disease status, age, sex, and other conditions, I found especially relevant the comparison between cross-sectional values and longitudinal rates of changes. The manuscript is concise and well-written, and the methods sections are clear. In that sense, I want to congratulate the authors for this great work.

I only have some suggestions with the aim of improving the manuscript:

Introduction:

The introduction is concise and well-written. However, I found that the aim of the study should be more clearly specified in the last paragraph of this section.

Results:

Association between CSF and plasma NfL: I recommend adding p values or these correlations.

NfL trajectory over disease progression:

One of the most relevant findings in this work is the different trajectories between plasma and CSF NfL changes in symptomatic carriers. In this comparison, authors grouped symptomatic carriers in one group, but it would be interesting to know if it is the case in the whole disease or if there are differences between the early and latest stages of the disease. Can the authors provide more data in that sense?

Methods:

My only concern in the methodology is if linear mixed effects models (LMMs) are the best way to treat this kind of data. I am perfectly aware that LMMS are the most popular way to treat longitudinal data, but NfL trajectories are not expected to be linear. The authors included a cubic spline with knots as previously done in other papers, and it should be enough to prevent non-linearity. However, I wonder if other statistical models - as generalized additive models (GAM; see <https://doi.org/10.1002/sim.9505>) - might produce better models. Can the authors shed some light on my doubts?

Discussion:

"We also find that the rates of change in both CSF and plasma NfL had somewhat earlier detectable increases (at 15-20 years before EYO) compared with cross-sectional NfL (10-15 years

before EYO).”: I found this sentence not very concrete. Looking the Figure 2, it is not clear that the differences between the rate of change in CSF appear earlier than the cross-sectional NfL levels. Can the authors provide the first EYO point where the groups differed in their models?

Figure 2: In the manuscript, the authors mention the “first EYO point where groups (NC and MC) differed” that “was determined to be the first point where the 95% credible intervals around the differences distribution between NC and MC did not overlap 0.” It would be helpful to establish visual differences if this referenced point were drawn in these plots (as vertical dashed lines for example).

Figure 3: Why do the authors prefer to specify the p values of comparisons between groups in the figure caption instead of in the figure itself? The latter would be visually more pleasant for the readers.

Figure 4: Although specified in the caption, it would be nice to add a legend with the color's meaning in this figure (as done in the other figures).

Table 1 and Table S1: The number of plasma and CSF samples per group should be detailed.

Table S2 and S3: Please, specify the reference for the gender factor (male or female)

Signed:

Sergi Borrego Ecija , MD, PhD

Reviewer #2 (Remarks to the Author):

In this study by Hofmann et al, they examine concentrations of NfL in plasma and CSF in the DIAN cohort in mutation and non-mutation carriers. The size (567 recruited to DIAN with baseline samples available: CSF n=442; plasma n= 507) and quality of the clinical phenotyping of the DIAN cohort makes it unique and extremely valuable. They are able to correlate plasma and CSF with atrophy rates and amyloid PET, and importantly, since they have access to longitudinal plasma and CSF biomarkers in N=349 . They manage to obtain very high numbers of longitudinal CSF (and plasma) samples which is testament to the organisation of this cohort and the motivation of the participants.

They show that plasma and CSF NfL are early biomarkers of neurodegeneration and that they are closely correlated in early disease. This is an important finding as we enter the phase of primary prevention treatment trials, where early and sensitive biomarkers of neurodegeneration are critical for success. This finding, particularly in plasma, makes this paper worthy of publication in its own right, in my opinion.

One of other major findings of the study is that the correlation between plasma and CSF is lost in more advanced disease. This is important to know, particularly in clinical trials where treatment trials are currently focussing (certainly in sporadic disease). However, I am slightly confused by the data that supports this statement. Examining Figure 3, it seems that the change in plasma NfL between symptomatic mutation carriers and converters is significantly different. I accept that the relationship is stronger for CSF but I think they need to amend this conclusion: “NfL appears to plateau between converter and symptomatic MC stages (Figure 3A), there was a step wise increase in the within-person rate of change in CSF NfL from presymptomatic MC to converter to symptomatic MC stages (Figure 3B).” In Figure 2, the change in plasma NfL does trend towards a less steep incline in plasma in the symptomatic group, but it does continue to climb.

The authors discuss reasons for this very briefly, but it would be interesting to discuss this further, particularly in the context of my comments above.

This reviewer is not surprised to see the uncoupling between NfL trajectory and amyloid deposition, but good correlations with atrophy rates. This substantiates earlier work which they reference appropriately.

Finally, they may have sufficient sample size to look at some of these relationships between different mutations. This reviewer would be very interested to see this data.

Specific Comments:

Figure 3: Could they show significance/p values on the figure itself – it is currently quite difficult to work out what is/ is not significant.

Figure S2: Is the Y axis mislabelled in Figure S2D – the legend refers to CSF but the figure is labelled plasma.

Conclusion: the sentence “maybe rendering CSF NfL the better read-out parameter in symptomatic stages of AD” is in my opinion too strong based on the data shown.

Reviewer #3 (Remarks to the Author):

In this study, Hofmann and collaborators assess the trajectories of neurofilament light chain protein (NfL) in cerebrospinal fluid (CSF) and plasma in a large, multicenter, well-characterized cohort of patients with autosomal dominant Alzheimer's disease (AD). The topic is of interest as these measures could be useful markers to track progression in clinical trials or in the use of disease-modifying treatments.

Main comments:

1. One of the main findings reported in the manuscript is that "While the step-wise between-group increase in the within-person rate of change in plasma NfL appears to plateau between converter and symptomatic MC stages (Figure 3A), there was a step-wise increase in the within-person rate of change in CSF NfL from presymptomatic MC to converter to symptomatic MC stages (Figure 3B)." However, according to the Figure 3 legend, "Sym MC had significantly higher rates of change compared to NC (B(s.e.m.) = 0.020 (0.002), $P < 2 \times 10^{-16}$), presym MC (B(s.e.m.) = 0.015 (0.002), $P = 4.55 \times 10^{-12}$), and converters (B(s.e.m.) = 0.003 (0.003), $P = 0.346$)." Please clarify whether the difference between Sym MC and converters is significant or not ($P = 0.346$?). If it is not, please correct the sentence in the legend. If it is, why do the authors conclude that within-person change in plasma NfL plateaus between Converter and Sym MC stages?

2. In Figure 3C, the authors show the ratio of plasma to CSF NfL concentrations. But the legend does not specify whether there are significant differences between Sym MC and converters, which would be the most relevant finding to support the assumption of a step-wise increase in CSF and a plateau in plasma. Could this be clarified?

3. The authors also report that "We also find that the rates of change in both CSF and plasma NfL had somewhat earlier detectable increases (at 15-20 years before EYO) compared with cross-sectional NfL (10-15 years before EYO)." This seems clear for plasma NfL, at least as shown in Figure 2, but it looks not so evident for CSF measures, where trajectories of both baseline levels and within-person rates of change seem to diverge equally at around EYO -15 / -16 (Figs 2B and 2D). Would it be possible to make this clearer by defining the specific EYO where trajectories diverge?

4. There seem to be discrepancies in the estimated EYO where trajectories diverge as shown in Figure 2 and those in Figure S1 (panels A, B, G & H). In Figure S1, the differences between MC and NC are significant (95% equal-tailed credible intervals not overlapping 0) before EYO -20 in all cases. Could the authors clarify the explanation for this discrepancy and which of the measures should be taken as more reliable? In this regard, adding a visual element to identify specific time points where 95% credible intervals around the difference distribution between NC and MC did not overlap would enhance the reader's understanding.

5. In the Discussion section, the authors state: "Overall, the underlying biological mechanism for these different dynamics of NfL in CSF and plasma in symptomatic phases of AD are not well understood but will be important to further investigate for determining which biofluid biomarkers to incorporate into clinical trials within this population." What do the authors suggest as next steps to further investigate these findings?

6. Please state the strengths and potential limitations of the study in the Discussion section.

Additional Considerations for Discussion:

1. The authors explain that a decrease in the ratio of plasma NfL/CSF NfL within symptomatic MC can be due to a clearing mechanism of NfL from brain to blood with disease progression. Could the authors provide a reference about this explanation? Otherwise, we recommend the authors

explicitly state that this point is a hypothesis.

2. The authors introduce an explanation about changes in the serum/CSF ratio of NfL in Guillain-Barré syndrome. However, is this pathology comparable to the subjects included in the present study? Guillain-Barré syndrome involves pathological processes affecting the CNS, and the elevation of CSF proteins without a corresponding increase in CSF cytology supports this fact. Based on these differences, the authors should reconsider including Guillain-Barré syndrome in their discussion. Could the authors clarify this statement? If not, we recommend avoiding this explanation in the paper.

3. Table S2: The results suggest an association of sex/gender with CSF NfL, but not with plasma NfL. Is this expected? What is the explanation for this discrepancy? Also, results shown in Table S3 suggest a sex/gender effect in baseline plasma NfL in the MC group, but this is not in line with results shown in Table S2. Could the authors provide a more detailed explanation regarding these findings?

4. Previous studies have shown that plasma NfL concentrations can be affected by renal function. It would be helpful to assess (or discuss) whether this variable could interfere with the results in this particular study.

5. In line 365, the authors explain that individuals with extreme values (n=3) for CSF and plasma NfL were reviewed and excluded from the analyses, specifying that "disc prolapse" was one of the reasons. Could the authors explain this relationship or provide a reference that justifies this exclusion? What were the other two reasons?

Minor comments:

1. Please include neuroimaging information (MRI and amyloid PET), both availability and descriptive statistics, in Table 1.

2. The reader would benefit from seeing the statistical results and between-group comparisons in the Figures instead of the legends, where they are hard to follow. Please consider including this information visually and shortening the legends.

3. It would be beneficial to indicate which was the reference category for sex/gender in Table S2, as this could help the reader better understand the data.

4. In line 866, regarding Figure 4: The authors mention that "similar associations were found when hippocampal atrophy was considered." Do the authors present these results? If not, please, specify this in the manuscript.

5. Please provide a definition for all acronyms the first time they are used (BMI, LMEM, LMM...) and use terminology consistently throughout the manuscript.

6. Please review terminology regarding sex (biological) and gender (social construct) and ensure consistency throughout the manuscript.

7. Line 125: Authors indicate that "An increase in NfL in blood and CSF is observed in AD but also in many other neurodegenerative diseases". Please note that high concentration of NfL in blood can also be found in non-neurodegenerative diseases.

8. Line 126: Authors use the word "enter" when they refer to NfL moving to CSF and blood. Perhaps other terms such as "diffuse" or "shift" would be more suitable.

9. For Figures 3 and 4, we encourage the authors to use titles that describe the purpose of the plot, rather than their results.

10. Line 838: Please correct the typo in "convertors".

Reviewer #4 (Remarks to the Author):

I co-reviewed this manuscript with one of the reviewers who provided the listed reports as part of the Nature Communications initiative to facilitate training in peer review and appropriate recognition for co-reviewers.

RESPONSE TO REVIEWER COMMENTS

We thank the Editor and Reviewers for their efforts in considering our manuscript thus far and appreciate the opportunity to submit a revised version of our attached manuscript for your consideration.

We provide point-by-point responses to all comments below as well as highlight all the changes made within the main text and supplemental materials in response to the Editor's and Reviewers' suggestions.

Reviewer #1 (Remarks to the Author):

This work uses data from the DIAN-Obs cohort to establish associations between plasma and CSF NfL levels across the Alzheimer's disease continuum. They focus on the comparison in the NfL trajectories between fluids (plasma Vs CSF) to elucidate if there are differences between them, information that can be useful for future clinical trials. Of note, a large number of longitudinal samples are included in this study. As some variability in the NfL levels between subjects is still not explained due to the disease status, age, sex, and other conditions, I found especially relevant the comparison between cross-sectional values and longitudinal rates of changes. The manuscript is concise and well-written, and the methods sections are clear. In that sense, I want to congratulate the authors for this great work.

I only have some suggestions with the aim of improving the manuscript:

Introduction:

The introduction is concise and well-written. However, I found that the aim of the study should be more clearly specified in the last paragraph of this section.

Thank you for this overall nice assessment of our work. We have now revised the last paragraph to more clearly specify our overall aim of our study and say "... [to] perform comparative analyses of the trajectories of CSF and blood NfL in relation to MRI volume and A β -PET changes in a large autosomal dominant AD sample and examine the distinct features of plasma and CSF NfL across the ADAD continuum." We also say in the second last paragraph of the introduction: "...is important for understanding the dynamics and significance of this fluid biomarker and to decide whether the more cumbersome and patient-burdening CSF measurement can be replaced by blood measurements".

Results:

Association between CSF and plasma NfL: I recommend adding p values or these correlations.

We have now added the corresponding p-values to the results section on page 5.

NfL trajectory over disease progression:

One of the most relevant findings in this work is the different trajectories between plasma and CSF NfL changes in symptomatic carriers. In this comparison, authors grouped symptomatic carriers in one group, but it would be interesting to know if it is the case in the whole disease or if there are differences between the early and latest stages of the disease. Can the authors provide more data in that sense?

We agree with the Reviewer that this is an interesting and important point. From data already presented in the manuscript, though perhaps not initially adequately discussed, we evaluate trajectories of CSF and plasma NfL across the entire disease spectrum (by using EYO as a proxy for disease time as a continuous measure). These data are visually represented in Figures 2, S1, and S2. Visually, we can see in Figures 2, S1, and S2 the rate of change in plasma NfL appears to slow at later EYOs (i.e., later disease stages) compared to the rate of change in CSF NfL. This is further supported by evaluating the piece-wise regression, presented in Figure S2, showing a single bifurcation point for rate of change in plasma NfL occurring at an EYO of 1.86 for MC group (Figure S2 panel C), depicting the trajectory of plasma NfL appears to plateau or even decrease after this point (also supported by the GAM modeling added to this new version of the manuscript as requested by this referee, Figure S2E-H). While, comparatively, there appears to be a steady linear increase in rate of change in CSF NfL across the entire disease course (i.e., no bifurcation point observed at later EYOs). We now more thoroughly describe this finding in the result section on page 6 and address this finding in the discussion section (page 8) by saying, “Overall the CSF and plasma NfL trajectories were found to be very similar at presymptomatic stages and discriminated MC from NC at 10-20 years before EYO dependent on the modeling used. Importantly, however, when examining rate of change in NfL over the disease course (EYO) and according to cognitive status, the rate of change in plasma NfL appears to plateau at later disease stages in symptomatic MC around two years after estimated symptom onset whereas the rate of change in CSF NfL tends to continue to increase even within symptomatic participants.”

Methods:

My only concern in the methodology is if linear mixed effects models (LMMs) are the best way to treat this kind of data. I am perfectly aware that LMMS are the most popular way to treat longitudinal data, but NfL trajectories are not expected to be linear. The authors included a cubic spline with knots as previously done in other papers, and it should be enough to prevent non-linearity. However, I wonder if other statistical models - as generalized additive models (GAM; see <https://doi.org/10.1002/sim.9505>) – might produce better models. Can the authors shed some light on my doubts?

In response to this referee comment we analyzed the NfL data using GAM. Overall, the curves seem similar to curves generated with LMEM in rstan and importantly highlight the unique curve for rate of change in plasma NfL, leveling off later in the disease. However, using the GAM method, we observed the divergence point between MC and NC is later compared to the initial LMEM method with restricted cubic spline and

mention this now in the abstract, i.e. that the divergence point is depending on the modeling. We suggest keeping the LMEM analyses in the main text and GAM in the supplement (now Figure S2E-H). First, because our initial work (Preisiche et al., 2019) was done with LMEM method and this manuscript was intended to be a follow-up. Moreover, the GAM method has a notable limitation (as mentioned in <https://doi.org/10.1002/sim.9505>) when applied to longitudinal data, that resulting smooths tend to have wider confidence intervals and less ability to discern differences in trends (in our case between MC and NC groups).

Discussion:

“We also find that the rates of change in both CSF and plasma NfL had somewhat earlier detectable increases (at 15-20 years before EYO) compared with cross-sectional NfL (10-15 years before EYO).”: I found this sentence not very concrete. Looking the Figure 2, it is not clear that the differences between the rate of change in CSF appear earlier than the cross-sectional NfL levels. Can the authors provide the first EYO point where the groups differed in their models?

We now provide additional details regarding the first EYO point where MC significantly differ from NC for cross-sectional and longitudinal plasma and CSF NfL trajectories and have added a dashed line in panels to indicate the EYO that MC and NC groups diverge. We do agree that overall the data may not be convincing enough to firmly conclude that rate of change diverges earlier than cross-sectional analysis and we do no longer mention this in the abstract as an overall conclusion.

We also rephrased the paragraph in the results part on page 6 that now reads: “Cross-sectionally, NfL levels in CSF and blood in the MC group began to increase, compared to NC group, between 15-25 years prior to expected symptom onset (Figure 2A, B with first difference between MC and NC observed at an EYO of -24.6 years for CSF and -18.9 years for plasma; Figure S1A, B). Similarly, when we examined within-person rate of change, we observed increases in NfL, compared to NC group, between 20-25 years prior to expected symptom onset (Figure 2 C, D with first difference between MC and NC observed at an EYO of -21.1 years for CSF and -21.2 years for plasma; Figure S1C, D). Supplementary analyses using a piece-wise regression method also resulted in similar trajectories with within-person rate of change diverging somewhat earlier compared to absolute levels (Figure S2). Moreover, a bifurcation point for rate of change in plasma NfL occurring at an EYO of +1.86 years for MC group was found (Figure S2C) where the trajectory of plasma NfL appears to plateau or even decrease after this point. While, comparatively, there appears to be a steady linear increase in rate of change in CSF NfL across the entire disease course (i.e., no bifurcation point observed at later EYOs; Figure S2D). An alternative generalized additive model (GAM) resulted in overall similar trajectories but a discrimination of MC vs NC was seen only at about 10 years prior to EYO (Figure S2E-H).”

Figure 2: In the manuscript, the authors mention the “first EYO point where groups (NC and MC) differed” that “was determined to be the first point where the 95% credible intervals around the differences distribution between NC and MC did not overlap 0.” It

would be helpful to establish visual differences if this referenced point were drawn in these plots (as vertical dashed lines for example).

As mentioned above, we now include a vertical dashed line in Supplemental Figure S1 (panels A-D) that corresponds to the first EYO where MC significantly differ from NC for cross-sectional and longitudinal plasma and CSF NfL trajectories.

Figure 3: Why do the authors prefer to specify the p values of comparisons between groups in the figure caption instead of in the figure itself? The latter would be visually more pleasant for the readers.

We have now added the corresponding p-values to the figures and removed reporting of statistics from figure legends. Further, we have moved the unstandardized beta estimates, standard errors, and corresponding p-values for between-group comparison to a new Table S4.

Figure 4: Although specified in the caption, it would be nice to add a legend with the color's meaning in this figure (as done in the other figures).

We have now revised the figures to better display a legend in the corresponding figure panels. Additionally, in response to Reviewers' comments we have moved the corresponding test statistics from the figure legend to a new Table S5.

Table 1 and Table S1: The number of plasma and CSF samples per group should be detailed.

We now report in the Table 1 figure legend that the overall N is 212 for NC group and 355 for MC, except for precuneus volume (NC = 200 and MC = 327), precuneus PiB-PET (NC = 186 and MC = 284), plasma NfL (NC = 193 and MC = 314), and CSF NfL (NC = 164 and MC = 278). We similarly report the corresponding number of individuals who have longitudinal CSF and plasma NfL samples available in NC and MC groups in Table S1.

Table S2 and S3: Please, specify the reference for the gender factor (male or female)

We now specify in the methods section and in the Tables that sex was self-reported as male or female and that the female group is the reference group in relevant analyses.

Reviewer #2 (Remarks to the Author):

In this study by Hofmann et al, they examine concentrations of NfL in plasma and CSF in the DIAN cohort in mutation and non-mutation carriers. The size (567 recruited to DIAN with baseline samples available: CSF n=442; plasma n= 507) and quality of the

clinical phenotyping of the DIAN cohort makes it unique and extremely valuable. They are able to correlate plasma and CSF with atrophy rates and amyloid PET, and importantly, since they have access to longitudinal plasma and CSF biomarkers in N=349. They manage to obtain very high numbers of longitudinal CSF (and plasma) samples which is testament to the organisation of this cohort and the motivation of the participants.

They show that plasma and CSF NfL are early biomarkers of neurodegeneration and that they are closely correlated in early disease. This is an important finding as we enter the phase of primary prevention treatment trials, where early and sensitive biomarkers of neurodegeneration are critical for success. This finding, particularly in plasma, makes this paper worthy of publication in its own right, in my opinion.

One of other major findings of the study is that the correlation between plasma and CSF is lost in more advanced disease. This is important to know, particularly in clinical trials where treatment trials are currently focussing (certainly in sporadic disease). However, I am slightly confused by the data that supports this statement. Examining Figure 3, it seems that the change in plasma NfL between symptomatic mutation carriers and converters is significantly different. I accept that the relationship is stronger for CSF but I think they need to amend this conclusion: "NfL appears to plateau between converter and symptomatic MC stages (Figure 3A), there was a step wise increase in the within-person rate of change in CSF NfL from presymptomatic MC to converter to symptomatic MC stages (Figure 3B)." In Figure 2, the change in plasma NfL does trend towards a less steep incline in plasma in the symptomatic group, but it does continue to climb.

The authors discuss reasons for this very briefly, but it would be interesting to discuss this further, particularly in the context of my comments above.

Thank you very much for this important comment.

As mentioned in response to a similar comment by Reviewer #1, from data already presented in the manuscript, though perhaps not adequately discussed, we evaluate trajectories of CSF and plasma NfL across the entire disease spectrum (by using EYO as a proxy for disease time as a continuous measure). These data are visually represented in Figure 2 and Figure S1 and S2 in the supplementary material. Visually, we can see in Figures 2, S1, and S2 the rate of change in plasma NfL appears to slow at later EYOs (i.e., later disease stages) compared to the rate of change in CSF NfL. This is further supported by evaluating the piece-wise regression, presented in Figure S2 panels C and D, showing a single bifurcation point for rate of change in plasma NfL occurring at an EYO of 1.86 for MC group (Figure S2 panel C), depicting the trajectory of plasma NfL appears to plateau or even decrease after this point. While, comparatively, there appears to be a steady linear increase in rate of change in CSF NfL across the entire disease course (i.e., no bifurcation point observed at later EYOs). We now more thoroughly describe this finding in the result part on page 6 and address this finding in the discussion section (page 8) by saying. "Overall, the CSF and plasma NfL trajectories were found to be very similar at presymptomatic stages and discriminated MC from NC at 10-20 years before EYO dependent on the modeling

used. Importantly, however, when examining rate of change in NfL over the disease course (EYO) and according to cognitive status, the rate of change in plasma NfL appears to plateau at later disease stages in symptomatic MC around two years after estimated symptom onset whereas the rate of change in CSF NfL tends to continue to increase even within symptomatic participants.”

Furthermore, we recognize the way we originally reported the results of the between-group analyses presented in Figure 3 was confusing. We now report the multiple comparison corrected p values and corresponding beta estimates in a table format (Table S4) instead of in the text of the figure legend, and for simplicity indicate, within the figure itself, only whether a specific between-group comparison was significant. We hope this will be easier for the readers to examine these relationships and conclude that there was no significant difference between Converters and Sym MC groups on the rate of change in plasma NfL.

This reviewer is not surprised to see the uncoupling between NfL trajectory and amyloid deposition, but good correlations with atrophy rates. This substantiates earlier work which they reference appropriately.

Finally, they may have sufficient sample size to look at some of these relationships between different mutations. This reviewer would be very interested to see this data.

Thank you for this insightful suggestion and we agree with the Reviewer that it would be interesting to look further are the relationships within and between different mutations. However, the majority of pathogenic variants in this study sample are in *PSEN1*, with significantly fewer variants located in *APP* and *PSEN2* genes (see Table S1). The limited sample sizes within *APP* (n = 58) and *PSEN2* (n = 23) mutations, especially for MC in the symptomatic stages (*APP* (n= 9) and *PSEN2* (n=0)), preclude the suggested analyses in the current study. We hope as the DIAN study continues we can better characterize how these relationships differ between mutations and cognitive groups. This limitation and future direction have now been added to the discussion section.

Specific Comments:

Figure 3: Could they show significance/p values on the figure itself – it is currently quite difficult to work out what is/ is not significant.

As also mentioned in response to Reviewer #1, we have now added the corresponding p-values to the figures and removed reporting of statistics from figure legends.

Figure S2: Is the Y axis mislabelled in Figure S2D – the legend refers to CSF but the figure is labelled plasma.

This is now corrected.

Conclusion: the sentence “maybe rendering CSF NfL the better read-out parameter in

symptomatic stages of AD” is in my opinion too strong based on the data shown.

We now say, “However, our observations suggest that in later stages of the disease CSF NfL changes may reflect the dynamic and ongoing neurodegeneration better than plasma NfL, maybe rendering CSF NfL the better read-out parameter in symptomatic stages of ADAD, compared to plasma NfL.”

Reviewer #3 (Remarks to the Author):

In this study, Hofmann and collaborators assess the trajectories of neurofilament light chain protein (NfL) in cerebrospinal fluid (CSF) and plasma in a large, multicenter, well-characterized cohort of patients with autosomal dominant Alzheimer's disease (AD). The topic is of interest as these measures could be useful markers to track progression in clinical trials or in the use of disease-modifying treatments.

Main comments:

1. One of the main findings reported in the manuscript is that "While the step-wise between-group increase in the within-person rate of change in plasma NfL appears to plateau between converter and symptomatic MC stages (Figure 3A), there was a step-wise increase in the within-person rate of change in CSF NfL from presymptomatic MC to converter to symptomatic MC stages (Figure 3B)." However, according to the Figure 3 legend, "Sym MC had significantly higher rates of change compared to NC (B(s.e.m.) = 0.020 (0.002), $P < 2 \times 10^{-16}$), presym MC (B(s.e.m.) = 0.015 (0.002), $P = 4.55 \times 10^{-12}$), and converters (B(s.e.m.) = 0.003 (0.003), $P = 0.346$)." Please clarify whether the difference between Sym MC and converters is significant or not ($P = 0.346$?). If it is not, please correct the sentence in the legend. If it is, why do the authors conclude that within-person change in plasma NfL plateaus between Converter and Sym MC stages?

Thank you for bringing this to our attention. We now report the multiple comparison corrected p values and corresponding beta estimates in a table format (Table S4) instead of in the text of the figure legend, and for simplicity indicate, within the figure itself, only whether a specific between-group comparison was significant. We hope this will be easier for the readers to examine these relationships and conclude that there was no significant difference between Converters and Sym MC groups on the rate of change in plasma NfL.

2. In Figure 3C, the authors show the ratio of plasma to CSF NfL concentrations. But the legend does not specify whether there are significant differences between Sym MC and converters, which would be the most relevant finding to support the assumption of a step-wise increase in CSF and a plateau in plasma. Could this be clarified

Thank you for your helpful comment. As mentioned above we have restructured the presentation of our results to help make interpretation of our findings easier. After adjusting for multiple comparisons correction, there is no significant difference between the ratio of plasma to CSF NfL between Sym MC and Converters (Figure 3C and Table S4). However, there are significant differences between SymMC and PreSym MC and the lack of significance between Sym and Converters is likely due to the low n of the converters.

3. The authors also report that "We also find that the rates of change in both CSF and plasma NfL had somewhat earlier detectable increases (at 15-20 years before EYO) compared with cross-sectional NfL (10-15 years before EYO)." This seems clear for plasma NfL, at least as shown in Figure 2, but it looks not so evident for CSF measures, where trajectories of both baseline levels and within-person rates of change seem to diverge equally at around EYO -15 / -16 (Figs 2B and 2D). Would it be possible to make this clearer by defining the specific EYO where trajectories diverge?

We now provide additional details in the text regarding the first EYO point where MC significantly differ from NC for cross-sectional and longitudinal plasma and CSF NfL trajectories. Additionally, we include a vertical dashed line in Supplemental Figure 1 (panels A-D) that corresponds to the first EYO where MC significantly differ from NC for cross-sectional and longitudinal plasma and CSF NfL trajectories. We also refer to Figure S2. We do agree that overall the data may not be convincing enough to firmly conclude that rate of change diverges earlier than cross-sectional analysis and we do no longer mention this in the abstract as an overall conclusion.

The paragraph in the results part on page 6 now reads: "Cross-sectionally, NfL levels in CSF and blood in the MC group began to increase, compared to NC group, between 15-25 years prior to expected symptom onset (Figure 2A, B with first difference between MC and NC observed at an EYO of -24.6 years for CSF and -18.9 years for plasma; Figure S1A, B). Similarly, when we examined within-person rate of change, we observed increases in NfL, compared to NC group, between 20-25 years prior to expected symptom onset (Figure 2 C, D with first difference between MC and NC observed at an EYO of -21.1 years for CSF and -21.2 years for plasma; Figure S1C, D). Supplementary analyses using a piece-wise regression method also resulted in similar trajectories with within-person rate of change diverging somewhat earlier compared to absolute levels (Figure S2). Moreover, a bifurcation point for rate of change in plasma NfL occurring at an EYO of +1.86 years for MC group was found (Figure S2C) where the trajectory of plasma NfL appears to plateau or even decrease after this point. While, comparatively, there appears to be a steady linear increase in rate of change in CSF NfL across the entire disease course (i.e., no bifurcation point observed at later EYOs; Figure S2D). An alternative generalized additive model (GAM) resulted in overall similar trajectories but a discrimination of MC vs NC was seen only at about 10 years prior to EYO (Figure S2E-H)."

4. There seem to be discrepancies in the estimated EYO where trajectories diverge as shown in Figure 2 and those in Figure S1 (panels A, B, G & H). In Figure S1, the

differences between MC and NC are significant (95% equal-tailed credible intervals not overlapping 0) before EYO -20 in all cases. Could the authors clarify the explanation for this discrepancy and which of the measures should be taken as more reliable? In this regard, adding a visual element to identify specific time points where 95% credible intervals around the difference distribution between NC and MC did not overlap would enhance the reader's understanding.

As mentioned above, we now include a vertical dashed line in Supplemental Figure 1 (panels A-D) that corresponds to the first EYO where MC significantly differ from NC for cross-sectional and longitudinal plasma and CSF NfL trajectories.

5. In the Discussion section, the authors state: "Overall, the underlying biological mechanism for these different dynamics of NfL in CSF and plasma in symptomatic phases of AD are not well understood but will be important to further investigate for determining which biofluid biomarkers to incorporate into clinical trials within this population." What do the authors suggest as next steps to further investigate these findings?

We now say on page 8: "Assays that allowing quantification of the central nervous system source vs peripheral nervous system source of NfL in body fluids would be very helpful as well as the assessment in renal function. Moreover, it will be important to do similar analyses in sporadic AD".

6. Please state the strengths and potential limitations of the study in the Discussion section.

We now provide such a paragraph at the end of the discussion (second last paragraph).

Additional Considerations for Discussion:

1. The authors explain that a decrease in the ratio of plasma NfL/CSF NfL within symptomatic MC can be due to a clearing mechanism of NfL from brain to blood with disease progression. Could the authors provide a reference about this explanation? Otherwise, we recommend the authors explicitly state that this point is a hypothesis.

We now say "may indicate a hypothetical change... "

2. The authors introduce an explanation about changes in the serum/CSF ratio of NfL in Guillain-Barré syndrome. However, is this pathology comparable to the subjects included in the present study? Guillain-Barré syndrome involves pathological processes affecting the CNS, and the elevation of CSF proteins without a corresponding increase in CSF cytology supports this fact. Based on these differences, the authors should reconsider including Guillain-Barré syndrome in their discussion. Could the authors clarify this statement? If not, we recommend avoiding this explanation in the paper.

We do not really understand this question and why a comparison with Guillain-Barré syndrome is not appropriate. The Reviewer says “Guillain-Barré syndrome involves pathological processes affecting the CNS, and the elevation of CSF proteins without a corresponding increase in CSF cytology supports this fact”; but is this not also true for AD pathogenesis.

3. Table S2: The results suggest an association of sex/gender with CSF NfL, but not with plasma NfL. Is this expected? What is the explanation for this discrepancy? Also, results shown in Table S3 suggest a sex/gender effect in baseline plasma NfL in the MC group, but this is not in line with results shown in Table S2. Could the authors provide a more detailed explanation regarding these findings?

We thank the Reviewer for their thoughtful comment. Our findings showing a sex effect on CSF NfL that has also been previously reported (Skillbäck et al. *Alzheimer’s and Dementia*, 2021). One potential explanation for the lack of sex effect on plasma NfL is the competing shared variance with BMI, which has a large, established (Manouchehrinia et al, *Ann Clin Transl Neurol*. 2020) effect on plasma measures of NfL, and our finding is support by prior work in larger studies demonstrating no effect of sex on blood NfL in healthy populations when age- and BMI-corrected (Kessler C et al. *Ann Clin Transl Neurol*. 2021 and Khalil et al. *Nat Commun*. 2020).

Regarding the apparent sex effect shown in Table S3 for MC: In this model, the main terms of interest are baseline \log_{10} CSF NfL or $\Delta\log_{10}$ CSF NfL. The interpretation of the effect of sex on plasma NfL in this model is in the context of CSF NfL as a covariate, which may lead to misinterpretation (i.e., “what is the effect of sex on plasma NfL, after adjusting for CSF NfL”) and is not intended by the authors to be interpreted in this way.

4. Previous studies have shown that plasma NfL concentrations can be affected by renal function. It would be helpful to assess (or discuss) whether this variable could interfere with the results in this particular study.

We agree with the Reviewer that understanding how plasma NfL concentrations may be impacted by renal function is an important point warranting discussion. Unfortunately, measures of renal function (e.g., creatinine levels) are not available in this study cohort. However, we now include this limitation in the discussion section on page 8.

5. In line 365, the authors explain that individuals with extreme values ($n=3$) for CSF and plasma NfL were reviewed and excluded from the analyses, specifying that "disc prolapse" was one of the reasons. Could the authors explain this relationship or provide a reference that justifies this exclusion? What were the other two reasons?

NfL increases after nerve tissue injury are documented in the literature (e.g. PMID:10222523) and are consistent of an increase of NfL after with any active disease of the central or peripheral nervous system (e.g. stroke, multiple sclerosis, polyneuropathy, ...) (PMID:30967444). In order to protect the anonymity of participants

and to maintain blinding of participants and study staff, we feel that it is inappropriate to give the exact neural medical causes for these few individuals. In this regard we also now delete “disc prolapse” and simply say: “Previous medical histories revealed that these three individuals had competing neurological disorders and were excluded from analyses (to maintain blinding the specific causes are not mentioned here)”.

Minor comments:

1. Please include neuroimaging information (MRI and amyloid PET), both availability and descriptive statistics, in Table 1.

Thank you for this comment. We have now added Mean (SD) and N for MC and NC groups for MRI and PiB-PET in the Table 1 legend.

2. The reader would benefit from seeing the statistical results and between-group comparisons in the Figures instead of the legends, where they are hard to follow. Please consider including this information visually and shortening the legends.

We have now added the corresponding p-values to the figures and removed reporting of statistics from all figure legends.

3. It would be beneficial to indicate which was the reference category for sex/gender in Table S2, as this could help the reader better understand the data.

We now specify that females are the reference group for the sex factor in the method and Table description.

4. In line 866, regarding Figure 4: The authors mention that “similar associations were found when hippocampal atrophy was considered.” Do the authors present these results? If not, please, specify this in the manuscript.

Thank you for bringing this to our attention. While atrophy in the precuneus is one of the earliest regions impacted in ADAD, hippocampal atrophy is perhaps a more classic region impacted early in sporadic AD and therefore initially included this brief statement regarding a similar association with hippocampal atrophy (please see figure below). However, the authors would like to highlight a comparison between the association between rate of change in NfL and grey-matter atrophy and the observed lack of association with amyloid-PET. The precuneus region is best suited for this comparison and therefore the authors have opted to just present the main results with the precuneus.

5. Please provide a definition for all acronyms the first time they are used (BMI, LMEM, LMM...) and use terminology consistently throughout the manuscript.

All acronyms are now defined the first time they are used.

6. Please review terminology regarding sex (biological) and gender (social construct) and ensure consistency throughout the manuscript.

We now specify in the methods section that sex was self-reported as male or female and that the female group is the reference group. We now use the term sex consistently throughout the manuscript.

7. Line 125: Authors indicate that "An increase in NfL in blood and CSF is observed in AD but also in many other neurodegenerative diseases". Please note that high concentration of NfL in blood can also be found in non-neurodegenerative diseases.

This statement has been revised to, "An increase in NfL in blood and CSF is observed in AD but also in many other neurodegenerative diseases (Bacioglu et al., 2016; Mattsson et al., 2019; Mielke et al., 2019) and some non-neurodegenerative diseases ((De Marchis et al., 2018; Korley et al., 2019)."

8. Line 126: Authors use the word "enter" when they refer to NfL moving to CSF and blood. Perhaps other terms such as "diffuse" or "shift" would be more suitable.

We have revised this sentence to say, "It is assumed that NfL is released from damaged neurons and then diffuses into the CSF and blood, serving as a biologic marker of neuronal damage."

9. For Figures 3 and 4, we encourage the authors to use titles that describe the purpose of the plot, rather than their results.

We have now changed the titles of Figure 3 and 4 to, “Association between rate of change in NfL and AD clinical groupings” and “Longitudinal association between NfL, brain atrophy, and amyloid deposition”.

10. Line 838: Please correct the typo in “convertors”.

Convertors has been corrected to converters.

Reviewer #4 (Remarks to the Author):

I co-reviewed this manuscript with one of the reviewers who provided the listed reports as part of the Nature Communications initiative to facilitate training in peer review and appropriate recognition for co-reviewers.

We thank this Reviewer for their co-review of our manuscript.

REVIEWERS' COMMENTS

Reviewer #1 (Remarks to the Author):

The authors have been very responsive to the previous comments and the manuscript is improved. My congratulations again for the work.

Reviewer #2 (Remarks to the Author):

The authors have adequately addressed my comments and concerns. I believe this should be published. It is an important addition to the field; high quality science in a unique cohort.

Reviewer #3 (Remarks to the Author):

I appreciate the responses provided by the authors. The revised version of the manuscript has improved substantially, and most of my concerns have been successfully addressed.

As a minor comment to avoid confusion for the reader, I suggest reconsidering the comparison to Guillain-Barré Syndrome. There are substantial differences between Guillain-Barré Syndrome and ADAD in terms of pathophysiology (inflammatory vs. neurodegenerative), topography (intrathecal and extrathecal vs. only intrathecal), and the chronopathology of the disease (a few weeks vs. decades). Although I am not entirely opposed to keeping it in the manuscript, using this example can be confusing for the reader and is not particularly helpful, especially when the authors have already justified their findings by referencing another, more similar, inherited neurodegenerative disease. However, it is ultimately the authors' choice whether to retain this comparison in the manuscript or not.

Reviewer #4 (Remarks to the Author):

RESPONSE TO REVIEWER COMMENTS

We thank the Editor and Reviewers for reviewing our revised manuscript and providing encouraging comments.

We provide point-by-point responses to the remaining comments below as well in the attached completed Author's Checklist and have highlighted the changes made within the main text and supplemental materials in response to the Editor's and Reviewer's suggestions.

Reviewer #1 (Remarks to the Author):

The authors have been very responsive to the previous comments and the manuscript is improved. My congratulations again for the work.

Thank you very much. We found your previous comments very helpful and are happy to hear you feel the manuscript is improved.

Reviewer #2 (Remarks to the Author):

The authors have adequately addressed my comments and concerns. I believe this should be published. It is an important addition to the field; high quality science in a unique cohort.

Thank you for your encouraging words and support of our work.

Reviewer #3 (Remarks to the Author):

I appreciate the responses provided by the authors. The revised version of the manuscript has improved substantially, and most of my concerns have been successfully addressed.

Thank you for your review of our revised manuscript based on your insightful comments.

As a minor comment to avoid confusion for the reader, I suggest reconsidering the comparison to Guillain-Barré Syndrome. There are substantial differences between Guillain-Barré Syndrome and ADAD in terms of pathophysiology (inflammatory vs. neurodegenerative), topography (intrathecal and extrathecal vs. only intrathecal), and the chronopathology of the disease (a few weeks vs. decades). Although I am not entirely opposed to keeping it in the manuscript, using this example can be confusing for the reader and is not particularly helpful, especially when the authors have already justified their findings by referencing another, more similar, inherited neurodegenerative disease. However, it is ultimately the authors' choice whether to retain this comparison in the manuscript or not.

While we agree with the Reviewer that the pathophysiology and other disease characteristics of Guillain-Barré Syndrome vs ADAD differ substantially, we focus our point more on the complex relationship between CSF and plasma NfL. We draw on the recent work from this field to highlight that as disease progresses there is evidence of differing relationships between CSF and plasma NfL, with no clear mechanistic explanation, providing a potential important next step for the field. Therefore, we have opted to leave this sentence in the discussion.

Reviewer #4 (Remarks to the Author):

We thank the Reviewer for their co-review of our manuscript.